# qPCR Assay as a Tool for Examining Cotton Resistance to the Virus Complex Causing CLCuD: Yield Loss Inversely Correlates with Betasatellite, Not Virus, DNA Titer

**DOI:** 10.3390/plants12142645

**Published:** 2023-07-14

**Authors:** Zafar Iqbal, Muhammad Shafiq, Sajed Ali, Muhammad Arslan Mahmood, Hamid Anees Siddiqui, Imran Amin, Rob W. Briddon

**Affiliations:** 1Central Laboratories, King Faisal University, Al-Ahsa 31982, Saudi Arabia; 2Department of Biotechnology, University of Management and Technology, Sialkot Campus, Sialkot P.O. Box 51340, Pakistan; sajed.ali@skt.umt.edu.pk; 3Agricultural Biotechnology Division, National Institute for Biotechnology and Genetic Engineering, Faisalabad 38000, Pakistan; muhammadarslan5581@gmail.com (M.A.M.); imranamin1@yahoo.com (I.A.); rob.briddon@gmail.com (R.W.B.); 4Department of Biotechnology, University of Sialkot, Sialkot P.O. Box 51340, Pakistan; hamid_anees@hotmail.com

**Keywords:** national coordinated varietal trials (NCVT), seed cotton yield (SCY), cotton leaf curl disease (CLCuD), cotton leaf curl Khokhran virus—Burewala strain (CLCuKoV-Bur), cotton leaf curl Multan betasatellite (CLCuMuB), quantitative real-time PCR (qPCR)

## Abstract

Cotton leaf curl disease (CLCuD) is a significant constraint to the economies of Pakistan and India. The disease is caused by different begomoviruses (genus *Begomovirus*, family *Geminiviridae*) in association with a disease-specific betasatellite. However, another satellite-like molecule, alphasatellite, is occasionally found associated with this disease complex. A quantitative real-time PCR assay for the virus/satellite components causing CLCuD was used to investigate the performance of selected cotton varieties in the 2014–2015 National Coordinated Varietal Trials (NCVT) in Pakistan. The DNA levels of virus and satellites in cotton plants were determined for five cotton varieties across three geographic locations and compared with seed cotton yield (SCY) as a measure of the plant performance. The highest virus titer was detected in B-10 (0.972 ng·µg^−1^) from Vehari and the lowest in B-3 (0.006 ng·µg^−1^) from Faisalabad. Likewise, the highest alphasatellite titer was found in B-1 (0.055 ng·µg^−1^) from Vehari and the lowest in B-1 and B-2 (0.001 ng·µg^−1^) from Faisalabad. The highest betasatellite titer was found in B-23 (1.156 ng·µg^−1^) from Faisalabad and the lowest in B-12 (0.072 ng·µg^−1^) from Multan. Virus/satellite DNA levels, symptoms, and SCY were found to be highly variable between the varieties and between the locations. Nevertheless, statistical analysis of the results suggested that betasatellite DNA levels, rather than virus or alphasatellite DNA levels, were the important variable in plant performance, having an inverse relationship with SCY (−0.447). This quantitative assay will be useful in breeding programs for development of virus resistant plants and varietal trials, such as the NCVT, to select suitable varieties of cotton with mild (preferably no) symptoms and low (preferably no) virus/satellite. At present, no such molecular techniques are used in resistance breeding programs or varietal trials in Pakistan.

## 1. Introduction

Cotton (*Gossypium* spp.) is the world’s most important fiber-producing plant, providing fiber for the textile industry as well as a significant portion of animal feed and edible oil. In terms of production, Pakistan ranks among the top ten cotton-producing countries. Cotton is grown on 2.96 million hectares in Pakistan, with an average annual lint yield of 670 kg ha^−1^ [1]. In 2020, the production of cotton lint, cotton seeds, and cotton oil reached 10,201,551 Mg, 2,252,784 Mg, and 280,800 Mg, respectively [2]. Cotton contributes 60% of the foreign exchange earnings to the economy of Pakistan. Currently, cotton production is severely affected by cotton leaf curl disease (CLCuD). CLCuD is caused by monopartite, single-stranded (ss) DNA viruses of the genus *Begomovirus* (family *Geminiviridae*) and is transmitted by an insect vector, *Bemisia tabaci* (whitefly). It is frequently associated with a disease-specific DNA-satellite molecule, *cotton leaf curl Multan betasatellite* (CLCuMuB) [3]. Nonetheless, another DNA-satellite molecule referred to as *cotton leaf curl Multan alphasatellite* (CLCuMuA) is occasionally found associated with CLCuD [4]. The precise contribution of CLCuMuB makes to the virus complex is unclear although it is essential for the infectivity of the virus or for the induction of symptoms. 

Since the first discovery of CLCuD-associated begomovirus in 1960, southern Asia has experienced four stages of CLCuD progression: pre-epidemic, epidemic, resistance breaking, and post-resistance breaking. The pre-epidemic phase of CLCuD was first experienced in 1967 near the city of Multan, Pakistan, but remained a sporadic problem affecting only a few plants [5]. In 1988, at Moza Khokhran, near Multan city, a whole field of an imported cotton variety was severely affected by CLCuD, and this was the beginning of a first epidemic termed as the ‘Multan epidemic’ that spread to almost all cotton growing areas of Pakistan and later into the northwestern states of India. CLCuD was first reported in India near the border area of Sri Ganganagar in 1989, and it spread as an epidemic from 1994 to 1998 [6]. The first CLCuD epidemic involved multiple species of begomoviruses, the most important of which were *Cotton leaf curl Kokhran virus* (CLCuKoV) and *Cotton leaf curl Multan virus* (CLCuMuV) associated with CLCuMuB [7,8]. By the late 1990s, plant breeders, using conventional breeding and selection methods, introduced new varieties with resistance derived from the varieties CP-15/2, LRA-5166, and CIM-443 [9]. Widespread cultivation of resistant varieties stemmed the losses from CLCuD [10]. However, from 2001 onwards, all the resistant cotton varieties began to show CLCuD symptoms, initiating in the vicinity of Burewala city, district Vehari, Pakistan, and named as Burewala epidemic or resistance-breaking epidemic. The virus involved in the Burewala epidemic was *Cotton leaf curl Kokhran virus* Burewala strain (CLCuKoV-Bur), previously known as *Cotton leaf curl Burewala virus* (CLCuBuV), and was associated with a recombinant betasatellite, termed as *Cotton leaf curl Multan Betasatellite* Burewala strain (CLCuMuB^Bur^) [11]. This resistance-breaking begomovirus (CLCuKoV-Bur/CLCuMuB^Bur^) complex infects all the previously resistance and tolerant cotton varieties in Pakistan and India [3,12,13]. CLCuKoV-Bur was referred to as a “resistance-breaking” strain, and after 2015, a gradual decrease in the onset of this strain was witnessed. Meanwhile, from 2015 onward, scientists witnessed a paradigm shift from this resistance-breaking strain to the rebound of the earlier CLCuD-associated begomoviruses in Pakistan [14] and India [15]. The onset of the third epidemic of CLCuD was predicted [16], and confirmed by subsequent studies [17]. 

It is worth noting that, aside from the sporadic identification of other CLCuD-associated begomoviruses and a mastrevirus from cotton throughout all epidemics, only a single species of betasatellite, CLCuMuB, was found associated with all the CLCuD-associated begomoviruses. The CLCuMuB belongs to a newly established family, *Tolecusatellitidae* (genus *Betasatellite*), and it has a small genome of 1350 nt with a conserved region and adenine-rich region. The indispensability of CLCuMuB highlights its importance and pivotal role. CLCuMuB encodes a multifunctional βC1 protein that mediates all the functions of a betasatellite. The βC1 protein has a major role in virulence and symptoms determination as well as manipulate the host cellular functions such as autophagy, ubiquitination, and suppression of gene silencing to support the CLCuD infection [18,19]. On the other hand, alphasatellite (previously referred to as DNA1) are self-replicating satellites like DNA molecules [4] that belong family *Alphasatellitidae* [20]. Alphasatellites have a ~1.4 kb ssDNA genome and their replication is mediated by a single encoded protein called replication-associated protein (*Rep^Alpha^*); Although, alphasatellites are self-replicating molecules and helper begomovirus mediates their encapsidation and movement. Alphasatellites have not yet been assigned a specific function, and neither the development of symptoms of infection nor the manifestation of disease is dependent on them. Alphasatellite, on the contrary, reduces symptoms in plants infected with begomovirus-betasatellite complex by regulating the virus and/or betasatellite DNA titers [21,22]. The *Rep^Alpha^* of some alphasatellites can, however, suppress the host defense at the post-transcriptional gene silencing level [23,24].

Several lines that were initially identified as resistant to the CLCuKoV-Bur/CLCuMuB^Bur^ complex in a large collection of available germplasm have turned susceptible [25]. Currently, there are no commercial cotton varieties available to farmers with resistance to the CLCuKoV-Bur/CLCuMuB^Bur^ complex. A more concerted effort is being made to look for resistant cotton because local cotton varieties in Pakistan are extremely susceptible to CLCuD. However, a few promising cotton lines are under investigation [1]. In Pakistan, the National Coordinated Varietal Trials (NCVTs) are conducted by the Directorate of Agricultural Research under the supervision of the Pakistan Central Cotton Committee (PCCC); an apex national research organization conducting research on cotton. The trials are being conducted with the aim of evaluating promising breeding material/strains developed by cotton breeders from federal and provincial institutions, as well as the private sector for suitability under varied climatic and soil conditions of the differing ecological zones of the country. Since the trials provide extensive information on varietal performance, this was chosen as the source of plant materials to assess the usefulness of quantitative real-time PCR (qPCR) analysis as an aid in determining the best cotton varieties for cultivation. With the limited availability of consumables, it was impossible to perform a comprehensive analysis of all varieties across all sites of the NCVT. Among these, three sites and five cotton varieties were selected to assess the possible use of qPCR to provide a quantitative measure of the susceptibility of a cotton variety to the virus complex using CLCuD.

Cotton yield is quantified by seed cotton yield (SCY), lint yield, lint percentage, boll number per plant, boll weight, lint index, and seed index. The most important of these are SCY and lint yield; the former is the total weight of seed and fiber, which reflects the potential productivity contributing to cotton fiber, whereas the latter is only the weight of fiber that is directly related to the textile industry [26]. SCY was used as a measure of varietal performance. Clearly it would have been preferable to use cotton lint yield as a measure of varietal performance. Lint yield, particularly quality, is the major indicator of the price which is obtained for the crop. Unfortunately, the NCVT does not measure the lint yield for each variety at each location. Rather, lint yield is measured for each variety across the whole trial. SCY was the only available yield parameter which was taken for each variety at each location and thus, this had to be used here as the measure of plant performance. Whether this choice has any effect on the results is unclear, but it is desirable, in the future, to investigate the relationship between virus/satellite DNA levels and lint yield/lint quality.

This study was aimed to envision the potential of qPCR to assess the titers of CLCuD-associated begomovirus and their effect on SCY. The results demonstrated that qPCR is a useful technique for assessing the virus/satellite DNA levels and revealed a significant negative correlation between betasatellite titer and SCY. 

## 2. Results

### 2.1. Phenotype and Disease Severity Index

From the ten NCVT sites, three were chosen: the Central Cotton Research Institute (CCRI) in Multan, the Nuclear Institute for Agriculture and Biology (NIAB) in Faisalabad, and the Cotton Research Station (CRS) in Vehari (part of the AYUB Agricultural Research Institute). Across all three sampling sites, the five selected varieties showed symptoms characteristic of CLCuD, indicating that no variety was immune (Figure 1). A wide range of symptoms, ranging from disease severity scale 1 to 4, were observed on the plants [27]. None of the selected varieties showed very severe symptoms (disease severity rating 5), indicating that all possess some level of tolerance to the disease. However, there were differences in symptom severity for the same variety at different locations (Table 1). Collectively, symptom severity was higher in Multan and Faisalabad than in Vehari. Notably, plants in Vehari and Faisalabad had large enations, leaf-like structures, on the adaxial side of the leaves, whereas plants in Multan had multiple but small enations. 

### 2.2. Standard Curve and Quantification

The standard curves for virus and satellites were obtained through linear regression analysis of threshold cycles (Ct) values plotted against the input DNA amount. The results showed a linear relationship between the Ct and the amount of input DNA over six log units (Appendix A). In addition, a melt curve analysis for each sample showed a single peak, indicative of the amplification of a single product (Appendix A).

The results of the qPCR analysis of the NCVT plants are provided in Table 1 and summarized in Figure 2. Additionally, the qPCR results are shown relative to the SCY for each variety at each location. No amplicon of virus or DNA satellite was obtained in any of the quantification cases from healthy control plants, highlighting the qPCR’s high sensitivity. Overall, the virus and alphasatellite titers were high in Vehari and Multan but low in Faisalabad; with the exception of B-23, for which the virus titer was high in Faisalabad. The highest virus titer was detected in B-10 (0.972 ng·µg^−1^) from Vehari, followed by B-2 (0.876 ng·µg^−1^) and B-1 (0.528 ng·µg^−1^) from Multan. The least virus titer was found in B-3 (0.006 ng·µg^−1^) from Faisalabad (Figure 2). Likewise, the highest alphasatellite titer was found in the B-1 (0.055 ng·µg^−1^), followed by B-10 (0.052 ng·µg^−1^), and B-2 (0.045 ng·µg^−1^), and all were from Vehari. The samples collected from Faisalabad had the lowest alphasatellite titer, with the least alphasatellite titer (0.001 ng·µg^−1^) in B-1 and B-2. The betasatellite titer was overall low in Multan relative to Faisalabad and Vehari (Figure 2). The highest betasatellite titer was found in B-23 (1.156 ng·µg^−1^) from Faisalabad, while the betasatellite titer in B-1 and B-2 from Faisalabad and Vehari was the same (0.959 ng·µg^−1^). B-12 from Multan had the lowest betasatellite titer (0.072 ng·µg^−1^) (Figure 2). 

### 2.3. Correlation between Virus Components and SCY

Correlation between the DNA levels of the three components, virus, beta-, and alphasatellite, of the CLCuD complex and SCY showed a highly varying pattern at a significance level of 5% (Table 2). The begomovirus titer showed a positive, but non-significant (*p* ≤ 0.05) correlation with betasatellite titer, alphasatellite titer, and SCY. Alphasatellite titer showed a non-significant, negative correlation with SCY (*p* ≤ 0.05), whereas begomovirus and betasatellite titers showed a non-significant positive correlation. Notably, betasatellite titer showed a significant and negative correlation with SCY. 

### 2.4. Path Coefficient Analysis

Path coefficient analysis revealed that virus titer had a positive effect on SCY and a negative effect on symptom severity, but in conjunction with betasatellite titer, it had a negative effect on both SCY and symptom severity (Figure 3). Begomovirus titer with alphasatellite titer had a similar negative effect on both SCY and symptom severity but less than that of betasatellite titer. For alphasatellite titer, the path coefficient analysis showed a negative effect on both SCY and symptom severity. The greatest effect was for betasatellite titer, having a significant inverse effect on SCY but the less negative effect on symptoms severity. Surprisingly, the analysis demonstrated that virus, betasatellite, or alphasatellite titer had no significant effect on symptoms severity. This may be explained by tolerant plants being able to tolerate quite high virus/satellite titers whilst exhibiting relatively mild symptoms whereas highly susceptible plants exhibit quite severe symptoms despite a relatively low virus/satellite titer. This finding is in agreement with the earlier study investigating the relationship between CLCuD symptoms and virus/satellite titer [27]. 

## 3. Discussion

The NCVT is a yearly series of trials designed to identify the best cotton varieties for cultivation in various agro-ecological zones of Pakistan. In this program, a dozen cotton varieties developed by federal, provincial, and private organizations are tested across at least twenty sites. Due to the scarcity of consumables, it was impossible to conduct a comprehensive analysis of all varieties across all sites. Instead, three sites and ten cotton varieties were selected to assess the possible use of qPCR to provide a quantitative measure of the susceptibility of a cotton variety to the virus, beta- and alphasatellite causing CLCuD.

Here, SCY was used to assess varietal performance. Clearly, cotton lint yield would have been a better indicator of varietal performance. Cotton lint yield and quality are the most important indicators of crop price. Unfortunately, the NCVT does not measure the lint yield for each variety at each location. Rather, lint yield for each variety is measured across the entire trial. SCY was the only available measure for each variety at each location, so this had to be used here as the measure of plant performance. It is unclear whether this choice has any effect on the results, but it will be interesting, in the future, to decipher the relationship between virus/satellite DNA levels and lint yield.

Considerable variability was found in both SCY and virus/satellite titer in the study shown here. The variability of the SCY between the varieties in a specific location may be due to their different genetic makeup, so they have differing levels of resistance/tolerance to the CLCuD complex. The variability of the SCY for one variety at different locations may be due to environmental conditions. It may also be the case that different locations have different “strains/variants” of the begomovirus/betasatellite/alphasatellite, yielding a different intensity of infection. There may also be some abiotic stresses, such as water stress which can also cause variability in SCY as well as other biotic stresses which may differ between the locations. It is precisely for this reason that the NCVT are conducted at multiple locations.

The results obtained here suggest that it is the betasatellite level that has the greatest effect on cotton performance rather than virus or alphasatellite titer, or even virus and betasatellite titer. This may not be surprising, even though the betasatellite depends entirely on the helper begomovirus for its encapsidation, replication, and movement [7]. Betasatellites encode only a single protein, known as βC1, which is a symptom/pathogenicity determinant [28,29], a suppressor of adaptive resistance mediated by small RNAs (so called RNA interference; [30,31,32], alters the pattern of micro RNA expression (micro RNAs being important in controlling plant development and gene expression [33], may upregulate the level of virus in plants [34]; and may also be involved in the movement of virus in plants [35], binds with DNA/RNA [31], interacts with a variety of host factors [36,37], interacts with the CP of helper begomovirus, forms multimeric complexes [38,39], suppresses jasmonic acid responses (involved in resistance to insects) in plants [40], impairs alphasatellite maintenance by a begomovirus [41], and induces autophagy [42]. It is clear that βC1 plays an important part in virus infection and can be the determining factor in plant performance. Some evidence which supports this is already available. Infection of cotton with CLCuMuV (one of the viruses causing CLCuD pre-resistance breaking) in the absence of the betasatellite induced only mild symptoms in cotton not typical of CLCuD [43]. Typical symptoms of CLCuD were shown to require co-infection with CLCuMuB [7], and Qazi et al. [28] showed that only CLCuMuB βC1 was required to induce CLCuD symptoms in plants.

Alphasatellite titer, either alone or in combination with begomovirus titer, has a negative effect on both symptom severity and SCY. Although alphasatellites are not required for geminivirus infection, the advantage of having one is subtle. Despite this, the presence of alphasatellites suggests that these molecules confer a significant selective advantage. The selective advantage of having an alphasatellite may be the ability to modulate symptoms to appeal insect vector behavior and performance, allowing swift viral propagation [44,45]. Furthermore, the autonomous replicating ability of alphasatellites may be attributed to the negative impact they have on SCY, as they consume more host resources, factors, and machinery. 

It is evident that the performance of cotton under varying environmental conditions and CLCuD pressure is complex, hence the need for trials such as those conducted under the NCVT. Nevertheless, the qPCR method outlined here would provide an additional character, other than symptoms, to determine/quantify cotton response to the viruses causing the disease. This can be useful in the decision-making process on which cotton varieties to carry forward in the development pipeline for commercially acceptable, high yielding varieties. Put simply, in the absence of immunity it is desirable for a variety to show mild symptoms (preferably no symptoms) and to harbor a low virus titer (preferably no virus), noting that the results obtained here suggest that it is betasatellite titer that is important rather than virus. So, the present NCVT trials monitor disease severity and yield parameters. qPCR would provide an additional, important parameter: virus/satellite titer. The authorities conducting the NCVT, and similar trials of cotton varieties, should provide serious consideration to including a qPCR element in the evidence for deciding which varieties to recommend farmers to grow. 

The qPCR assay demonstrated here has successfully been implemented to quantify the titers of different begomoviruses and their associated DNA satellites in pumpkin plants [46], sunflower plants [47], and tomato plants (M. Shafiq unpublished data). The results here suggest that the qPCR can be simplified to measure betasatellite levels only, thus reducing complexity and cost. Additionally, multiplexing qPCR to detect all components of CLCuD, using specialized probes (such as TaqMan probes) shortens the time and lowers the cost. The system then improves the selection of varieties for commercial cultivation, benefiting both farmers and the country. The finding may also be of significance for other diseases caused by begomovirus/betasatellite complexes, suggesting that the betasatellite is more a factor in crop losses than the virus. This possibility requires investigation.

In this study, only one attribute, SCY, was investigated to measure cotton yield. Nonetheless, other attributes such as lint yield, total biomass, lint percentage, and boll number per plant, can formulate a broader perspective and can be included in futuristic studies. Betasatellite performs several key functions and βC1 induces virus-like symptoms [28], which ultimately affect plant phenotype, biomass, and leaf area, thus severely undermining the foundation of high cotton yield (seed and lint) [26,48]. However, no study entails the proteomic profile of cottonseed infected with begomovirus. Such studies follow the example set by He et al. [49], who compared distribution of storage protein in glanded and glandless cottonseeds. These issues are the focus of future studies. 

## 4. Materials and Methods

### 4.1. Data Collection and DNA Extraction

Ten leaf samples from ten different plants of the same variety were collected from three of the ten sites of the 2014–2015 NCVT trials; Nuclear Institute for Agriculture and Biology (NIAB) Faisalabad (31°39′13″ N; 73°04′18″ E), the Central Cotton Research Institute (CCRI) Multan (30°14′03″ N; 71°42′41″ E), and the Cotton Research Station (CRS) Vehari (30°03′35″ N; 72°36′05″ E)(part of the AYUB Agricultural Research Institute). The inclusion of the five cotton varieties (designated as B-01, B-02, B-10, B-12, and B-23) in the study was based on their susceptibility to CLCuD. SCY data from 100 plants of each variety were collected from three of ten sites of the 2014–2015 NCVT trials. Plants were photographed prior to the leaf collection (Figure 1), and collected leaves were labelled, and transported to the lab on icebox.

DNA was extracted from the collected leaves and the control healthy cotton plants (grown in greenhouse conditions) by the CTAB method [50]. The resulting DNA pellet was dissolved in sterile distilled water (SDW) and stored at −20 °C. The concentration of isolated genomic DNA from cotton leaves was measured using a NanoDrop spectrophotometer (ND-2000, Bio-Rad, Hercules, CA, USA).

### 4.2. Design of Primers

Oligonucleotide primers for qPCR were designed to alignments of sequences of CLCuD-associated begomoviruses and their associated satellites downloaded from the nucleotide sequence database. ClustalX2 was used to align the retrieved sequences, and areas of conservation were identified to which primers were designed and synthesized as described in [27]. BegomoqPCRF1/BegomoqPCRR1 primers were used to quantify the begomovirus titer; for betasatellite, BetaqPCRF2/BetaqPCRR2 primers were used; and for alphasatellite, AlphaqPCRF2/AlphaqPCRR2 primers were used (Table 3) and were demonstrated earlier [16]. All DNA samples used for qPCR quantification were first normalized against *Sad1* (acc. No. AJ132636; [51]) values and then used for qPCR quantification.

### 4.3. Preparation of Standard Curves for Quantification and Conditions for qPCR

Standards for the absolute quantification and condition for qPCR for the identification of CLCuD associated with begomoviruses and their associated satellites were described previously [27].

A total of 25 µL were used for each reaction, of which half was SYBER Green Super Mix (Thermo Fisher Scientific, Waltham, MA, USA). Thermocycler (iQ5 thermal cycler, Bio-Rad, Hercules, CA, USA) was used, and the final settings were demonstrated earlier [16]. In qPCR, the extracted DNA from the leaves of healthy cotton plants was used as a negative control and standards were used as positive controls. The extracted DNA from the leaves of the same variety from the same location was pooled and each sample was tested three times. 

### 4.4. Data Analysis

Then qPCR was performed with begomoviruses and their cognate satellites under the same conditions used in [17]. Each template data set was analyzed independently using iQTM5 Optical System software (version 2.1, Bio-Rad, Hercules, CA, USA), and all melt and standard curves attributes, as well as Ct values, were also inferred. 

To assess the correlation structure between the titers of CLCuD components and SCY, PASW Statistics 18 (formerly SPSS Statistics; SPSS Inc., Chicago, IL, USA (http://www.spss.com.hk/statistics/; accessed on 20 January 2017) was used at a significance level of 5%. Path coefficient analysis was performed at a significance level of 5% to assess the correlation of combinations of components of the CLCuD complex SCY by using the software IBM SPSS Amos graphics (http://www-03.ibm.com/software/products/en/spss-amos; accessed on 4 February 2017).

## 5. Conclusions

Conclusively, the qPCR assay was found to be a useful tool in examining the response of cotton plants to the virus complex that causes CLCuD, identifying resistant sources against CLCuD in breeding programs, and monitoring the spread and population of CLCuD-associated viruses in epidemiological surveys. As the titers of begomovirus and alphasatellite varied greatly, no relationship was found between the titers of begomovirus and alphasatellite to SCY was concluded. Nonetheless, a statistically significant and inverse relationship between betasatellite titer and SCY was noted.

Differences in yield at different locations could be accredited to differences in environmental conditions and, in the case of viruses and satellites, differences in virus/satellite “strains” between locations.

## Figures and Tables

**Figure 1 plants-12-02645-f001:**
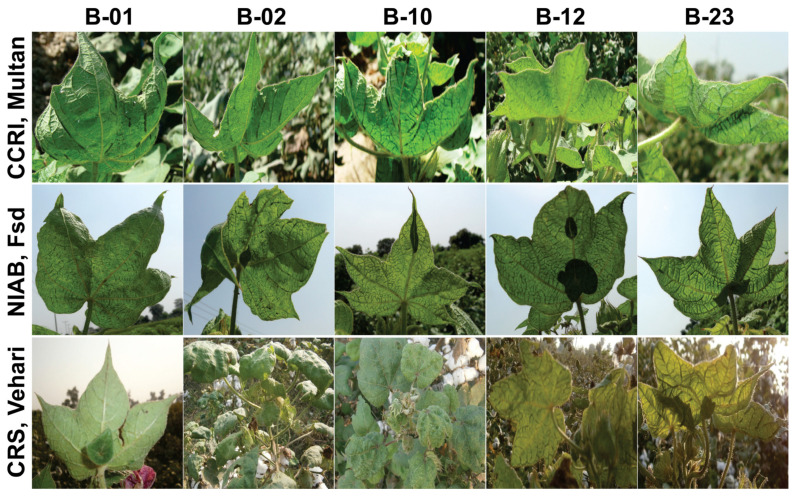
Symptoms exhibited by the field collected samples for the varieties B-01, B-02, B-10, B-12, and B-23 of the 2014–2015 NCVT trials in the Central Cotton Research Institute (CCRI) Multan, National Institute for Agriculture and Biology (NIAB) Faisalabad (Fsd), and Cotton Research Station (CRS) Vehari. Plants B-12 at CCRI Multan, and B-01 and B-02 at NIAB Fsd were showing symptoms of vein darkening/swelling at Scale-1. Plants B-01, B-02, B-10, and B-23 at CCRI Multan were exhibiting Scale-2 symptoms, characterized by vein darkening/swelling, leaf curling, and enation. Plants B-10, B-12, and B-23 at NIAB Fsd, and B-01 at CRS Vehari were showing Scale-3 phenotype, which included vein darkening/swelling, leaf curling, leaf crumpling, enation, and stunting. Plants B-02, B-10, B-12, and B-23 at CRS Vehari were showing Scale-4 symptoms, characterized by vein darkening/swelling, leaf curling, leaf crumpling, enation, severe stunting, and extensive crumpling.

**Figure 2 plants-12-02645-f002:**
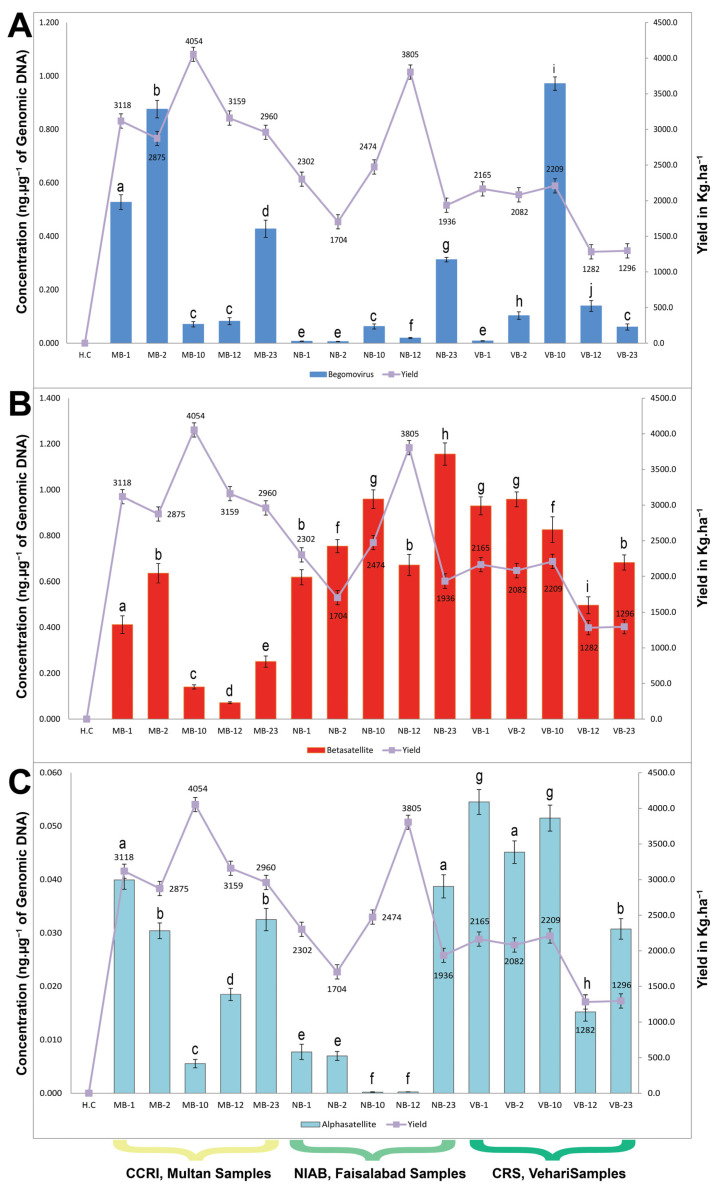
qPCR estimation of the virus, betasatellite, and alphasatellite titers (ng·µg^−1^) in total DNA extracted from the NCVT trials cotton plants. The scatter chart displays the SCY of the varieties at each location as well as the calculated titer of begomovirus (**A**) betasatellite (**B**) and alphasatellite (**C**) with standard deviation (shown in vertical bars). Values preceded by the same letter are not significantly different at the 5% level using a two tailed t-test. The NCVT codes used for different varieties are B-1, B-2, B-10, B-12, and B-23; nonetheless, the prefix letter represents the location of the trial. The samples collected from CCRI Multan are prefixed with M, from NIAB Faisalabad with N, and from CRS Vehari with V. The healthy control (H.C) is a non-infected glasshouse-grown cotton plant. Values mentioned in the figure are mean of three independent replicates (*n* = 3) and values denoted by different letters (in small alphabets) differ significantly at 5% significance level.

**Figure 3 plants-12-02645-f003:**
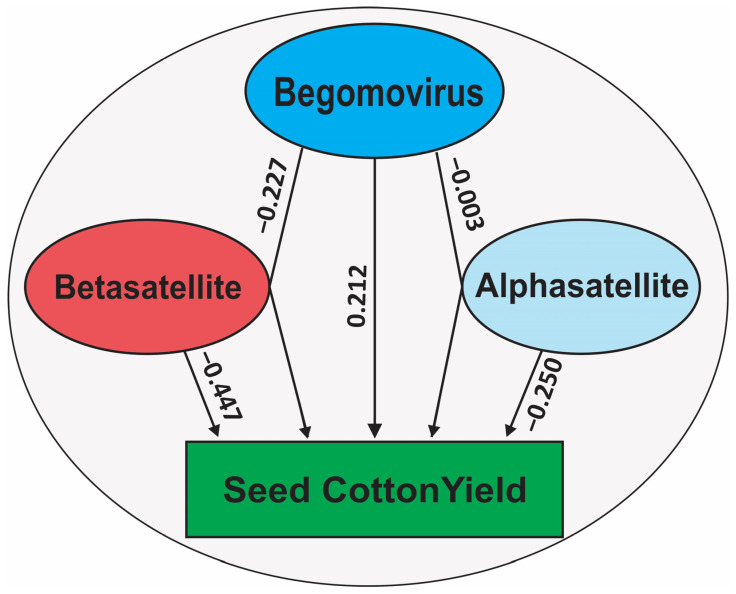
Path coefficient analysis of the correlation of virus, betasatellite and alphasatellite with SCY and symptom severity at a significance level of 5%.

**Table 1 plants-12-02645-t001:** Determined virus, betasatellite, and alphasatellite titers in 5 cotton varieties across three geographical locations.

Variety	CCRI, Multan					NIAB, Faisalabad					CRS, Vehari				
	Severity ^@^	Virus (ng·µg^−1^)	Beta. (ng·µg^−1^)	Alpha. (ng·µg^−1^)	SCY * (kg·ha^−1^)	Severity ^@^	Virus (ng·µg^−1^)	Beta. (ng·µg^−1^)	Alpha. (ng·µg^−1^)	SCY * (kg·ha^−1^)	Severity ^@^	Virus (ng·µg^−1^)	Beta. (ng·µg^−1^)	Alpha. (ng·µg^−1^)	SCY * (kg·ha^−1^)
B-01	3	0.528 (±0.028)	0.412 (±0.039)	0.040 (±0.002)	3118.0 (±100)	3	0.008 (±0.0008)	0.619 (±0.032)	0.008 (±0.001)	2302.0 (±100)	2	0.009 (±0.0008)	0.930 (±0.039)	0.055 (±0.002)	2165.0 (±100)
B-02	3	0.876 (±0.033)	0.636 (±0.043)	0.030 (±0.001)	2875.0 (±100)	2	0.006 (±0.0007)	0.754 (±0.029)	0.007 (±0.001)	1704.0 (±100)	3	0.103 (±0.0141)	0.959 (±0.032)	0.045 (±0.002)	2082.0 (±100)
B-10	2	0.071 (±0.010)	0.140 (±0.009)	0.006 (±0.001)	4054.0 (±100)	3	0.063 (±0.0096)	0.959 (±0.041)	0.00001 (±0.0001)	2474.0 (±100)	2	0.972 (±0.0253)	0.826 (±0.056)	0.052 (±0.002)	2209.0 (±100)
B-12	3	0.082 (±0.013)	0.072 (±0.004)	0.018 (±0.001)	3159.0 (±100)	2	0.020 (±0.0020)	0.672 (±0.046)	0.00001 (±0.0001)	3805.0 (±100)	3	0.139 (±0.0204)	0.497 (±0.037)	0.015 (±0.002)	1282.0 (±100)
B-23	2	0.428 (±0.032)	0.251 (±0.024)	0.033 (±0.002)	2960.0 (±100)	3	0.313 (±0.0086)	1.156 (±0.049)	0.039 (±0.002)	1936.0 (±100)	1	0.060 (±0.0115)	0.687 (±0.033)	0.031 (±0.002)	1296.0 (±100)

@ Disease severity rating. Disease severity is marked on a 5 point scale from severity level 0 (no symptoms) to severity level 4 (severe symptoms) as described in [27]. The value provided is the average of approx. 100 plants for each variety at each location (±standard deviation). * Seed cotton yield.

**Table 2 plants-12-02645-t002:** Analysis of the correlation of the titers of virus, betasatellite and alphasatellite with SCY and symptoms severity.

	SCY	Begomovirus	Betasatellite	Alphasatellite	Symptom Severity
SCY	1				
Begomovirus	0.083	1			
Betasatellite	−0.540 *	0.013	1		
Alphasatellite	−0.289	0.508	0.307	1	
Symptom severity	0.379	−0.292	−0.114	−0.327	1

* Correlation is significant at the 0.05 level.

**Table 3 plants-12-02645-t003:** Oligonucleotide primers used in the study.

Primer	Position	Sequence (5′-3′)	Amplicon Size (bp)
BegomoqPCRF1	V2	ATGTGGGATCCACTGTTAAATGAGTTCCC	186
BegomoqPCRR1		GATTATATCTGCTGGTCGCTTCGACATAA	
BetaqPCRF2	βC1	CAAGTATATCAAGTCTGTGAACTATATCTT	194
BetaqPCRR2		GATACTATCCACAAAGTCACCATCGCTAAT	
AlphaqPCRF2	Rep	ATTCAAATTTCAAATTTGAAATCTTGGCA	249
AlphaqPCRR2		CCTTCTTATCACGAGGGATATCAAATACAA	

## Data Availability

All the work related to this study is mentioned in the main script and Appendix A.

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
