# Peer review of "qPCR Assay as a Tool for Examining Cotton Resistance to the Virus Complex Causing CLCuD: Yield Loss Inversely Correlates with Betasatellite, Not Virus, DNA Titer"

_plants, 2023, doi:10.3390/plants12142645_

Round 1

Reviewer 1 Report

(1)The more results could be provided in the manuscript

(2) Suggest to rearrage Table 1, and put the brackets in the one line

(3) Add the specfic primer sequences in the manuscript

(4) Could the qPCR assay be suitable for the other plant species ? such as tomato

The quality of English language in this manuscript meets the criteria for publication.

Reviewer 2 Report

The topic of this manuscript is interesting to the readers of Plants. English writing is OK. It should be publishable after revision. .

Specific comments

Abstract. Please provide more numbers (quantitative values/data) if possible.

L42 and other places. Weight unit ”tonnes” should be expressed with metric symbol “Mg”.

LL42-43. Deleted “accessed on 42 April 25, 2023)”.

L123. Better to have a little more info on the term “seed cotton”, such as, “seed cotton (i.e., lint and seed)” or alike.  

LL135-135. The sentence seems not fitting well here at the end of the introduction.

L201. The unit (ng/μg.μg‒1) seems not right. Are those the SD or SE bars. What is the replicate (n=?).

L219. What is the sample number (n=?).

L237. Can you give the significance levels of these path coefficients?

L315. For a quality paper with a greater impact, better to offer some outlook on the future work at the end of your discussion section. For example, 1) referring to your argument on LL123-135, should you look for the correlationship of the begomovirus/betasatellite factors with other yield parameters (e. g,, lint, cottonseed, oil or total plant biomass); 2) As betasatellites encode multi-functional pathogenicity proteins, could proteometric analysis of the cottonseed also provide some insight into the pursuit (e. g., impacts on protein functions, types, and/or abundance)?  Such as protein profiling found some differences between two types of cottonseeds in functional peptides, including vicilin-like antimicrobial peptides (Agric Environ Lett. 2022;7:e20076. https://doi.org/10.1002/ael2.20076).

English writing is OK

Round 2

Reviewer 1 Report

(1)The resolution of the figures in this manuscript needs to be oncreased.

(2)The format of the table 1 and table 3 in this manuscript should be three-line form.
